# DHFR Inhibitors Display a Pleiotropic Anti-Viral Activity against SARS-CoV-2: Insights into the Mechanisms of Action

**DOI:** 10.3390/v15051128

**Published:** 2023-05-09

**Authors:** Daniela Iaconis, Francesca Caccuri, Candida Manelfi, Carmine Talarico, Antonella Bugatti, Federica Filippini, Alberto Zani, Rubina Novelli, Maria Kuzikov, Bernhard Ellinger, Philip Gribbon, Kristoffer Riecken, Francesca Esposito, Angela Corona, Enzo Tramontano, Andrea Rosario Beccari, Arnaldo Caruso, Marcello Allegretti

**Affiliations:** 1EXSCALATE, Dompé Farmaceutici SpA, Via Tommaso De Amicis, 95, 80131 Napoli, Italy; daniela.iaconis@dompe.com (D.I.);; 2Section of Microbiology Department of Molecular and Translational Medicine, University of Brescia, 25123 Brescia, Italy; francesca.caccuri@unibs.it (F.C.);; 3Dompè Famaceutici SpA, Via Campo di Pile snc, 67100 L’Aquila, Italy; 4Fraunhofer Institute for Translational Medicine and Pharmacology ITMP, Schnackenburgallee 114, 22525 Hamburg, Germany; 5Fraunhofer Cluster of Excellence for Immune-Mediated Diseases CIMD, Theodor-Stern-Kai 7, 60596 Frankfurt am Main, Germany; 6Research Department Cell and Gene Therapy, Department of Stem Cell Transplantation, University Medical Center Hamburg-Eppendorf, 20246 Hamburg, Germany; 7Department of Life and Environmental Sciences, University of Cagliari, Cittadella Universitaria SS554, 09042 Monserrato (CA), Italy

**Keywords:** COVID-19, drug repurposing, methotrexate, EXSCALATE, virtual screening, molecular docking, anti-viral activity, SARS-CoV-2, viral entry, nsp13

## Abstract

During the COVID-19 pandemic, drug repurposing represented an effective strategy to obtain quick answers to medical emergencies. Based on previous data on methotrexate (MTX), we evaluated the anti-viral activity of several DHFR inhibitors in two cell lines. We observed that this class of compounds showed a significant influence on the virus-induced cytopathic effect (CPE) partly attributed to the intrinsic anti-metabolic activity of these drugs, but also to a specific anti-viral function. To elucidate the molecular mechanisms, we took advantage of our EXSCALATE platform for in-silico molecular modelling and further validated the influence of these inhibitors on nsp13 and viral entry. Interestingly, pralatrexate and trimetrexate showed superior effects in counteracting the viral infection compared to other DHFR inhibitors. Our results indicate that their higher activity is due to their polypharmacological and pleiotropic profile. These compounds can thus potentially give a clinical advantage in the management of SARS-CoV-2 infection in patients already treated with this class of drugs.

## 1. Introduction

Since the beginning of the COVID-19 pandemic, tremendous efforts have been made by the scientific community to find therapeutic approaches for the treatment of the SARS-CoV-2-induced respiratory disease. In this ongoing research, the repurposing of approved drugs has been considered the most rapid, affordable and efficient strategy [1], and numerous available drugs have thus been tested in repositioning studies over the last two years. 

Among the tested drugs, methotrexate (MTX), a widely used chemotherapy and immunosuppressant drug [2,3,4,5,6,7,8,9], has shown anti-viral effects against SARS-CoV-2 [10,11]. Similar to other anti-folates, MTX exerts its anti-cancer function mainly by inhibiting the dihydrofolate reductase (DHFR), an enzyme involved in the de novo synthesis of the nucleosides required for nucleic acid production. This inhibition of DHFR results in anti-metabolic and anti-inflammatory effects due to the subsequent direct or indirect inhibition of several cellular mechanisms, such as lymphocytes replication, polyamine production, redox cellular activity and cytokines release [12] and [13,14,15,16,17,18,19]. These activities on inflammation as well as nucleic acid metabolism and purine synthesis suggest that MTX could be beneficial for COVID-19 [20]. Further, additional evidences showed a role of this drug in also regulating the activity of the Angiotensin Converting Enzyme 2 (ACE2) and the interaction of a human helicase with spike (S) and Transmembrane Serine Protease 2 (TMPRSS2) (https://opendata.ncats.nih.gov/covid19, access on 13 July 2022 [20,21,22]). As such, these data indicate that MTX may also interfere with viral entry and replication by targeting host proteins.

Belonging to the same drug class of MTX, pralatrexate (PTX) is another DHFR inhibitor used for the treatment of relapsed or refractory peripheral T-cell lymphoma (PTCL). Interestingly, an effect in counteracting COVID-19 disease has also been observed for this compound [23], thus supporting the hypothesis that other DHFR inhibitors may have anti-viral effects against SARS-CoV-2. 

Based on from these data, in this study, we evaluated the anti-viral activity of several DHFR inhibitors, specifically PTX, trimetrexate (TMX), aminopterin, as well as MTX, pemetrexed and raltitrexed, against SARS-CoV-2. We then assessed the effects of these compounds in inhibiting virus entry and, taking advantage of our EXSCALATE platform for molecular docking simulations, we also examined other targets and mechanisms potentially involved in the inhibition of SARS-CoV-2 infection mediated by DHFR inhibitors that could be explained by the different anti-viral activities of these compounds. 

## 2. Materials and Methods

### 2.1. Cells

An african green monkey kidney Vero E6 cell line was obtained from the American Type Culture Collection (ATCC) and maintained in Dulbecco’s Modified Eagle’s Medium (DMEM; Gibco, Thermo Fisher Scientific, Waltham, MA, USA) supplemented with 10% fetal bovine serum (FBS; Gibco, Thermo-Fisher Scientific, Waltham, MA, USA) at 37 °C in a humidified atmosphere of 5% CO_2_ [24]. A human lung epithelial carcinoma cell line A549, overexpressing angiotensin-converting enzyme 2 (ACE2), A549 ACE2+ cells, a kind gift of Prof. Steven J. Elledge (Harvard Medical School, Boston, MA, USA), were maintained in DMEM supplemented with 10% FBS at 37 °C in a humidified atmosphere of 5% CO2.

### 2.2. Virus

We successfully isolated SARS-CoV-2 in Vero E6 cells from the nasopharyngeal swab sample of a COVID-19 patient. The identity of the strain was verified in Vero E6 cells by real-time polymerase chain reaction (PCR) and by metagenomic sequencing, from which the reads mapped to nCoV-2019 [genomic data are available at Global Initiative on Sharing All Influenza Data (GISAID) under the accession n. EPI_ISL_1379197]. We propagated the clinical isolate in Vero E6 cells and determined the viral titer by a standard plaque assay. Infections were carried out using the SARS-CoV-2 B.1 lineage [11,25] at a multiplicity of infection (MOI) of 0.05. All the infection experiments were performed in a biosafety level-3 (BLS-3) laboratory.

### 2.3. Cell Viability Studies of Compounds

Cells were seeded into 24-well plates (2.5 × 10^4^ cells/well) in DMEM supplemented with 10% FBS and treated with the indicated doses of each compound at 37 °C for 48 h. Cell viability was evaluated by measuring the ATP levels using CellTiter-Glo (Promega, Madison, WI, USA). 

### 2.4. Evaluation of Anti-Viral Efficacy of Compounds

Cells were infected at 37 °C for 1 h with SARS-CoV-2 at a MOI of 0.05. Infections were carried out in DMEM without FBS. Then, the virus was removed, and cells washed with a warm phosphate buffered saline (PBS) and cultured with a medium containing 2% FBS in the presence or in the absence of different concentrations of each compound. Cells and supernatants were collected for further analysis at 48 h post infection (p.i). 

### 2.5. Plaque Assay

Cells were seeded at a density of 5 × 10^5^ cells/well in a 12-well plate and incubated at 37 °C for 24 h. Supernatants from infected cells were serially diluted in DMEM without FBS and added to the cells. After a 1 h incubation, media were removed, and the cells were washed with a warm PBS. Then, cells were covered with an overlay consisting of DMEM with 0.4% SeaPlaque (Lonza, Basel, Switzerland). The plates were further incubated at 37 °C for 48 h. Cells were fixed with 10% formaldehyde at room temperature for 3 h. Formaldehyde was aspirated, and the agarose overlay was removed. Cells were then stained with crystal violet (1% CV *w*/*v* in a 20% ethanol solution), and viral titer (PFU/ml) of SARS-CoV-2 was determined by counting the number of plaques.

### 2.6. Viral RNA Extraction and Quantitative Real-Time RT-PCR (qRT-PCR)

RNA was extracted from clarified cell culture supernatants (16,000 g × 10 min) and from infected cells using the QIAamp Viral RNA Mini Kit and RNeasy Plus mini kit (Qiagen, Hilden, Germany), respectively, according to the manufacturer’s instructions.

RNA was eluted in 30 μL of RNase-free water and stored at −80 °C until use. The qRT-PCR was carried out following previously described procedures [26]. Briefly, reverse transcription and amplification of the S gene were performed using the one-step QuantiFast Sybr Green RT-PCR mix (Qiagen) as follows: 50 °C for 10 min, 95 °C for 5 min; 95 °C for 10 s, 60 °C for 30 s (40 cycles) (primers: RBD-qF1: 5′-CAATGGTTTAACAGGCACAGG-3′ and RBD-qR1: 5′-CTCAAGTGTCTGTGGATCACG-3). A standard curve was generated by determination of copy numbers derived from serial dilutions (10^3^–10^9^ copies) of a pGEM T-easy vector (Promega, Madison, WI, USA) containing the receptor binding domain of the S gene (primers: RBD-F: 5′-GCTGGATCCCCTAATATTACAAACTTGTGCC-3′; RBD-R: 5′-TGCCTCGAGCTCAAGTGTCTGTGGATCAC-3′). 

### 2.7. Western Blot Analysis

A western blot was carried out following previously described procedures with minor modifications [27]. Protein samples (30 µg) obtained from lysis in the RIPA buffer (Cell Signaling Technology, Danvers, MA, USA) of infected cells were separated on 10% SDS-PAGE and then transferred onto polyvinylidene difluoride (PVDF) membranes (Millipore, Sigma, Burlington, MA, USA). After being blocked with 3% BSA in a TBS buffer containing 0.05% Tween20, the blot was probed with a human serum (1:1000 dilution) containing IgG to the SARS-CoV-2 nucleoprotein (NP) and with mouse anti-human GAPDH monoclonal antibody (G-9: Santa Cruz Biotechnology, Dallas, TX, USA). The antigen-antibody complexes were detected using peroxidase-conjugated goat anti-human or goat anti-mouse IgG (Sigma) and revealed using the enhanced chemiluminescence (ECL) system (Santa Cruz Biotechnology, Dallas, TX, USA).

### 2.8. Data Analysis

The half-cytotoxic concentration (CC50) and the half-maximal inhibitory concentration (IC50) for each compound were calculated from concentration-effect-curves after non-linear regression analysis using Prism8 Software (GraphPad Software, La Jolla, CA, USA). The selectivity index (SI) was calculated as the ratio of CC50 over IC50 [28].

### 2.9. Statistical Analysis

Data were analyzed for statistical significance using the 1-way ANOVA, and the Bonferroni post hoc test was used to compare data. Differences were considered significant when *p* < 0.05. Statistical tests were performed using Prism8 Software (GraphPad Software, San Diego, CA, USA).

### 2.10. Pseudovirus Entry Assay

Cell-free lentiviral particles were generated as described previously [29] and production protocols are available at the LeGO website (http://www.LentiGO-Vectors.de, access on 13 July 2022). Lentiviral particles (2.9 × 10^5^ TU/mL) used for this study contain the D614G variant of spike without the last 19 amino acids to remove the ER-retention signal. Caco-2 cells, obtained from Cell Lines Service (CLS, #300137), were grown in DMEM High Glucose (4.5 g/l) (Capricorn, #DMEM-HXRXA) with 10% fetal bovine serum (Capricorn, #FBS-12A), L-Glutamine (Capricorn, #GLN-B), streptomycin (100 μg/mL) and 100 U/mL penicillin (Capricorn, #PS-B). Cells were seeded in 20 µL at a density of 8000 cells per well into white 384-well microtiter plates (Greiner Bio-One, #781073) and incubated at 37 °C in the presence of 5% CO2 for 24 h. Compounds were added using Echo550 (Labcyte Inc., San Jose, CA, USA) directly prior to virus addition. Virus addition was done in 10 µL/well and incubated for 48 h at 37°C. Detection was conducted by the addition of a 30 µL of 0,5 mM Luciferin (Biosynth Carbosynth; #FL08608) solution in PBS and incubation at 10 min in the dark at RT and measurement of Luminescence using the EnSight multimode plate reader with 100 ms detection time per well. Data analysis was performed using Microsoft Excel and GraphPad Prism 8. Test compound results were normalized relative to the DMSO control that represents 0% inhibition of lentiviral pseudovirus entry. Dose response curves were fitted to 4-parameter logistic functions in Prism8 Software (GraphPad Software) with no constraints.

### 2.11. Nsp13 Enzymatic Assay

SARS-CoV-2 nsp13 was expressed from the pNIC-ZB vector (Addgene plasmid #159614; RRID:Addgene_159614) in Rosetta cells, using a TB medium for culture and purified according to Newman et al. [30]. SARS-CoV-2 nsp13 enzymatic activity was evaluated as reported [31]. Briefly, nsp13 unwinding-associated activity was measured in black 384-well plates (PerkinElmer), in 40 μL reaction volume containing 20 mM Tris–HCl, pH 7.2, 50 mM NaCl, 2 μM Hel Capture oligo (5′-TGG TGC TCG AAC AGT GAC-3′) from Biomers, 5 mM MgCl2, 5% DMSO or inhibitor and 1 nM of purified nsp13. The reaction mixture containing the enzyme was pre-incubated for 10 min with an inhibitor at room temperature (RT). The reaction was started by adding 1 mM ATP and 750 nM annealed DNA substrate (5′-AGT CTT CTC CTG GTG CTC GAA CAG TGA C-Cy3-3′, 5′-BHQ-2-GTC ACT GTT CGA GCA CCA CCT CTT CTG A-3′) from Biomers. After 15 min of incubation at 37 °C, products were measured with Victor Nivo (Perkin) at 530/580 nm.

### 2.12. SARS-CoV-2 nsp12 RNA Polymerase RNA Dependent (RdRp) Activity

SARS-CoV-2 nsp12 was expressed from a pET28a vector in BL21 DE3 cells, as previously described [32]. Briefly, the protein was firstly purified with a Ni-Sepharose column and eluted in a buffer containing 20 mM Tris pH8, 150 mM NaCl, 1M Imidazole and 4 mM MgCl2. The fraction containing the protein was loaded in a HiTrapQ-HP column and eluted in a buffer containing 20 mM Tris pH8, 1M NaCl and 4 mM MgCl2. The quality of the protein was analyzed through SDS-PAGE, and the purified protein was stored at -80°C. The SARS-CoV-2 nsp12 RdRp activity was measured in black 96-well plates (PerkinElmer), in 25 µL reaction volume containing 50 mM Tris-HCl pH 8.0, 1 mM DTT, 2.5 mM MgCl2, 50 mM NaCl, 10% Glicerolo, 20 µM UTP, RNAse inhibitor 10 U/µL (Thermo Scientific), 12.5 nM polyA RNA template, 4% DMSO or inhibitor and 400 nM of purified nsp12. The reaction mixture containing the enzyme and the template RNA was incubated for 60 min with the inhibitor at 37°C. After the incubation, the reaction was stopped with the addition of 2 µL of 200 mM nuclease-free EDTA. Then, 170 µL of 1× PicoGreen (Invitrogen) in 1× TE were added to the mixture and the reaction was incubated for 5 min at RT, protected from light. Products were measured with Victor Nivo (PerkinElmer, Waltham, Massachusetts, United States) at 502/523 (em/ex) nm. Experiments were performed in triplicate; the results report the average and standard deviation of two independent replicates.

## 3. Results

### 3.1. Inhibitors of DHFR Exert Anti-Viral Activity against SARS-CoV-2 in Two Cell Lines

Previous studies have indicated an MTX-mediated anti-viral activity against SARS-CoV-2 infection, possibly due to MTX effects on host cellular processes ([11,20,21] and patent: WO 2022049275, WO 2022106489). To analyze the potential anti-viral activity of other DHFR inhibitors, we first performed a classical cellular phenotypic assay on Vero E6 cells, as previously described [24]. This cell line has been extensively used for SARS-CoV-like virus studies and is highly susceptible to cell death after infection [33,34,35,36,37,38]. We used cells that constitutively expressed the EGFP fluorescent protein, which allows for monitoring the effects of drug treatment in modulating the virus-induced cytopathic effect (CPE) by measuring EGFP fluorescence. In parallel, cytotoxicity was determined in the absence of the virus to establish the half-maximal cytotoxic concentration (CC50) for each compound. Although cells did not reach 100% of confluence after treatment due to the anti-metabolic activity of this class of compounds, the observed half-maximal inhibitory concentrations (IC50) were much lower than the CC50 values, indicating good therapeutic windows (Figure 1). CPE analysis showed that PTX, TMX and aminopterin have higher anti-viral activity in these cells, and quantitative PCR (qPCR) readouts evaluating the effects of compounds on viral replication were in line with these results (Figure 1 and [24]).

Additionally, we tested this class of compounds on A549 ACE2+ cells, a human lung epithelial cell line engineered to stably overexpress the ACE2 receptor [39,40]. A549 ACE2+ cells are commonly used for in vitro screening and characterization of drug candidates against SARS-CoV-2 and have already been used to study MTX cellular pathways [39,41,42,43], using different approaches. 

At first, a standard assay was carried out to measure the activity of each DHFR inhibitor on A549 ACE2+ cell metabolism and to determine their selectivity index (SI) (Table 1) [11,37]. To this end, cells were cultured for 48 h in the absence or presence of different drug concentrations. The cells treated with these compounds displayed a normal surface-adherent phenotype at all concentrations tested (Figure 2a, Figure 3a and Figure 4a), and the CC50 values for PTX and TMX were found to be 0.008 µM and 0.01 µM, respectively (Figure 2b, Figure 3b and Table 1). On the other hand, MTX and raltitrexed displayed a lower tolerability with CC50 values of 1.18 and 0.89 µM, respectively (Figure 4b and Table 1). Lastly, pemetrexed and aminopterin had the lowest tolerability with a CC50 higher than 2 µM (Table 1).

To assess the anti-viral activity of the compounds, A549 ACE2+ cells were then infected with SARS-CoV-2 for 1 h at a multiplicity of infection (MOI) of 0.05 [11] and, after infection, cells were cultured in the absence or presence of different concentrations of DHFR inhibitors. Supernatants were then collected at 48 h post infection (p.i.) and tested for viral genome copy numbers by quantitative real-time PCR (qRT-PCR). Three out of the six DHFR inhibitors tested showed a strong anti-viral effect, displaying a IC50 < 1 µM (Table 1). The most active compounds were PTX, TMX and MTX, which significantly reduced the virus yield displaying a dose-dependent inhibition of viral replication (Figure 2c, Figure 3c and Figure 4c). In particular, PTX exhibited a 90.4% to 96.5% inhibition of viral titer at 0.019 µM and 0.039 µM, respectively (Figure 2d). The efficacy of the treatment was also confirmed at the intracellular level by qRT-PCR and western blot (WB) on SARS-CoV-2 nucleoprotein (NP) (Figure 2e,f), with the IC50 value calculated to be 0.004 µM. At the same time, TMX significantly reduced the SARS-CoV-2 virus yield, with an 86% reduction at a concentration of 0.009 µM and a 97.5% and 96.5% inhibition at 0.019 µM and 0.039 µM, respectively (Figure 3d). The efficacy of the treatment was confirmed at the intracellular level by qRT-PCR and WB on NP (Figure 3e,f), with the IC50 value calculated to be 0.007 µM. Among the three compounds, MTX was the least efficient, with an IC50 value calculated to be 0.63 µM (Table 1). In particular, MTX exhibited a 94% to 98% inhibition of viral titer at 1.25 µM and 2.5 µM, respectively (Figure 4d). The efficacy of the treatment was also confirmed at the intracellular level by qRT-PCR and WB on SARS-CoV-2 NP (Figure 4e,f). Collectively, these data indicate that PTX, TMX, and MTX, exert high anti-viral activity in the low nanomolar range. On the contrary, the remaining three tested compounds, aminopterin, pemetrexed and raltitrexed, showed low anti-viral activity with IC50 values > 1 µM, and specifically of 1.3, 5 and 5 µM, respectively (Table 1). Taken together these data demonstrate that the anti-cytopathic effect of DHFR inhibitors is due not only to the anti-metabolic action of these compounds, but also to a specific anti-viral activity.

### 3.2. PTX, TMX and MTX Inhibit the Activity of SARS-CoV-2 Viral Key Enzymes

We then sought to understand whether the different anti-viral activities of the compounds could be attributed to different effects on viral proteins. To identify the potential targets of DHFR inhibitors among the viral proteins, we took advantage of the high-throughput screening campaign run in the context of the E4C consortium (https://www.exscalate4cov.eu/, access on 13 July 2022). We performed a virtual screening using our EXSCALATE platform for molecular docking simulations, as already described. The simulation was performed using LiGen™ (Ligand Generator), the de novo structure-based virtual screening software, designed and developed to run on HPC architectures, which represents the most relevant tool of the EXSCALATE platform (https://ieeexplore.ieee.org/document/9817028, access on 13 July 2022). From this analysis, we obtained the docking score values that predict the binding affinity of the molecules in the protein binding site, and based on this information, we tested the six compounds for their potential inhibitory effects against two important enzymes in mediating SARS-CoV-2 replication: nsp13 helicase, which is essential for viral replication [31], and RNA-dependent RNA polymerase (RdRp), which is a highly versatile enzyme that is involved in the RNA viral genome replication process. In line with data on anti-viral activity, PTX and TMX were the most effective in inhibiting the unwinding activity of the nsp13 helicase, displaying IC50 values of 0.14 and 1.56 µM, respectively, while MTX showed an IC50 of 2.03 µM. Compared to PTX, TMX and MTX, the other tested compounds were less effective in this assay, showing progressively higher IC50 values: 2.59 µM for aminopterin, 3.22 µM for raltitrexed and 10.83 µM for pemetrexed (Figure 5). Finally, none of the tested compounds were able to inhibit RdRp (IC50 > 30 µM for all the tested compounds).

Altogether, these data demonstrate that PTX, TMX and MTX have the highest anti-viral effect against SARS-CoV-2 due to a dual mechanism of action. Positive strand RNA viruses remodel cell metabolism to create a suitable microenvironment to survive and replicate in host cells. Indeed, the CPE observed in infected cells is ascribed to a viral hijacking of cellular resources to fulfill viral needs. Anti-metabolite drugs, aimed to subtract nucleotides required for the synthesis of viral RNA or impair protein translation, act as broad-spectrum anti-viral drugs. Thus, DHFR inhibitors, on one side, inhibit cell metabolism, and on the other side, they can inhibit the function of key viral enzymes, thus exhibiting a pleiotropic effect. On the contrary, pemetrexed, raltitrexed and aminopterin, which have shown a lower anti-viral effect (IC50 > 1.5 µM), do not affect viral replication mechanisms, suggesting that their activity is only due to the anti-metabolic effect of this drug class.

### 3.3. DHFR Inhibitors also Inhibit SARS-CoV-2 Viral Entry

Finally, because previous studies indicated that MTX can also inhibit viral entry by acting on virus–host interactions [21], we finally investigated whether these mechanisms could also be controlled by PTX and TMX. We used the pseudovirus entry assay to investigate the potential effects of PTX and TMX on viral entry. To this end, permissive Caco-2 cells naturally expressing ACE2 were infected with a SARS-CoV-2 pseudovirus. Notably, the pseudovirus used was a lentiviral vector pseudotyped expressing SARS-CoV-2 spike protein (D614G) on the surface and carrying a luciferase (Luc2) reporter.

Again, in line with the high anti-viral activity of these compounds, PTX, TMX and, confirming previous studies, also MTX significantly inhibited the luciferase output with IC50 values in the nanomolar range, further demonstrating that these drugs exert their anti-viral activity through a polypharmacological effect, as they can strongly interfere also with viral entry (Figure 6). Furthermore, the potential inhibitory activity of DHFR inhibitors on serine protease TMPRSS2, which is important for spike protein priming, was assessed. Only PTX displayed a significant inhibition of TMPRSS2, with an IC50 of 0.45 µM, and this effect was not observed with TMX and MTX (IC50 > 30 µM; Axxam S.p.A. proprietary assay, Bresso, Italy, 2021). Notably, PTX was the compound displaying the highest anti-viral activity and was also the one showing the highest and widest ability of targeting both viral proteins and mechanisms of viral entry, further demonstrating that the different anti-viral activities that we observed using different DHFR inhibitors depend on the pleiotropy of their mechanism of action.

## 4. Discussion

By combining in silico studies with experiments using human COVID-19 in vitro models, in this study, we demonstrated that in addition to MTX, PTX and TMX also possess strong anti-viral effects against SARS-CoV-2. The superior effects of these compounds relative to other DHFR inhibitors in counteracting the viral infection were due to polypharmacological activity, targeting not only the metabolism but also virus entry and other mechanisms involved in virus replication.

Although different studies have sought to explain the effects of MTX using different in vitro models of COVID-19 [11,20,21] and have suggested dual activity of the drug on both viral entry and virus replication in the host, to date, the underlying mechanisms of these effects have not been completely explained and targets have not been clearly identified. Taking advantage of our EXSCALATE platform for molecular docking simulations, we identified potential targets of MTX and also of PTX and TMX, and then experimentally demonstrated the inhibition of the nsp13 helicase as a new target for the mediation of the superior anti-viral activity of all three of these DHFR inhibitors and their influence on viral entry as a common mechanism underlying the anti-viral effects of these drugs. Moreover, we identified TMPRSS2 as an additional target protein inhibited by the most effective DHFR inhibitor, PTX.

Fighting the SARS-CoV-2-induced respiratory disease is still a top priority for the scientific community, and investigating and explaining the potential anti-viral effects of drugs, such as DHFR inhibitors, that are used by patients that would be highly exposed to severe complications in case of infection is of utmost importance. MTX is widely and successfully used for the treatment of cancer and autoimmune diseases, but its toxicity, poor pharmacokinetic and narrow safety range are certainly the major issues associated with its prolonged use [44,45] and often lead to dose reduction or withdrawal of treatment [8,46,47]. MTX bioavailability is relatively high, but it is highly bound to plasma proteins and shows a very low volume of distribution, suggesting that its biodistribution may not be sufficient to reach the primary target tissues of the lung, while accumulating instead in the liver and intestine [13,48]). PTX and TMX are known, respectively, as anti-cancer and anti-opportunistic infection agents [49,50,51]). PDX has a more effective biodistribution ([52] and https://www.drugbank.com/, access on 13 July 2022), showed greater in vitro and in vivo anti-tumor efficacy [53,54]) and gave promising results in in vivo studies and clinical trials about toxicity, indicating a safer profile compared to MTX [52,53,55,56]. In addition, PTX is under study for the treatment of Non-Small Cell Lung Cancer (NSCLC), suggesting that it can reach the lung more efficiently compared to MTX. On the other hand, TMX is the least studied drug among DHFR inhibitors. Differently from the above-mentioned drugs, it targets DHFR specifically but not the thymidylate synthase, suggesting that toxicity concerns typical of the other members of the same family could be overcome in this case. Interestingly, TMX also shows an indication for lung fungal infections [57] and was reported to be distributed into the respiratory tract [58].

Our studies demonstrated that PTX and TMX have strong anti-viral efficacy against SARS-CoV-2 and that their higher anti-viral activity (nanomolar range) compared to the other compounds belonging to this drug class is due to their polypharmacological profile and pleiotropic effects. The anti-metabolic activity observed for the most effective DHFR inhibitors only partially explain the anti-viral activity of these compounds, and a direct role on viral entry and replication mechanisms significantly contribute to the anti-CPE activity of these compounds, resulting in an additive anti-viral effect. Thus, additionally considering their pharmacokinetic features, PTX and TMX could be even more effective than MTX in the management of SARS-CoV-2 infection-associated complications in patients affected by chronic diseases who are already using these drugs.

The first clinical evidence of the potential beneficial effects of the treatment with a DHFR inhibitor in the COVID-19 disease was observed in patients receiving MTX for treating psoriasis or rheumatoid arthritis [59,60,61,62,63,64,65]. These studies suggested that MTX treatment did not worsen COVID-19 outcomes or rates of hospitalization in these patients, probably due to effects on inflammation associated with the SARS-CoV-2 infection [66,67,68]; however, as this cohort of patients was heterogeneous in terms of the period of MTX treatments and clinical manifestations of COVID-19, no clear conclusion on the benefits of MTX therapies could be extrapolated. Similarly, further supporting the potential beneficial effect of MTX in COVID-19 patients’ treatment, the Global Rheumatology Alliance physician-reported registry recently reviewed the COVID-19 situation in rheumatic patients, suggesting that odds of death were higher in patients receiving a different Disease-Modifying Anti-Rheumatic Drug (DMARD) or not receiving any DMARD compared with patients treated with MTX alone [69].

Awareness of the strong, pleiotropic anti-viral activity of PTX and TMX may be extremely useful for physicians managing SARS-CoV-2 infections in patients with cancers or autoimmune diseases treated with these drugs. These patients are in fact often immunocompromised, cannot undergo vaccination or are at higher risk of insufficient immune response after vaccines and of developing severe COVID-19 diseases [70]. In this context, treatment with DHFR inhibitors such as PTX and TMX with strong anti-viral activity and better safety profiles may be advantageous, allowing for treatment of the primary disease and, eventually, controlling the COVID-19 disease.

## 5. Conclusions

With this study, we confirm the importance of repurposing studies and of in silico/experimental synergy as very powerful methods to generate effective responses to diseases that are still untreatable. Using this approach, we demonstrated that PTX, TMX and MTX have stronger anti-viral effects against SARS-CoV-2 compared to other DHFR inhibitors, and such higher efficacy is due to their pleiotropic inhibitory activity on cell metabolism as well as both viral entry and replication mechanisms. These compounds can potentially provide a clinical advantage in the management of SARS-CoV-2 infection-associated complications in patients affected by chronic diseases who are already treated with this class of drugs.

## Figures and Tables

**Figure 1 viruses-15-01128-f001:**
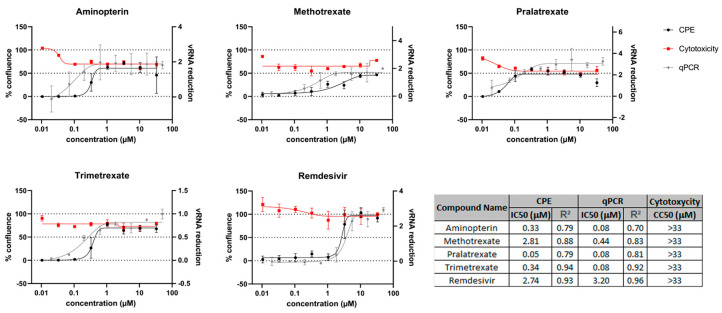
DHFR inhibitors screened for anti-viral activity on Vero E6 cell line. The anti-viral activity was evaluated by measuring the % of cell confluence at different drug concentrations. CPE analysis and quantitative PCR (qPCR) readouts were used to calculate the IC50 (black and gray curves, respectively). Red lines in the graph show the antimetabolic effect of the DHFR inhibitor in the absence of SARS-CoV-2 infection. Remdesivir was used as a positive control.

**Figure 2 viruses-15-01128-f002:**
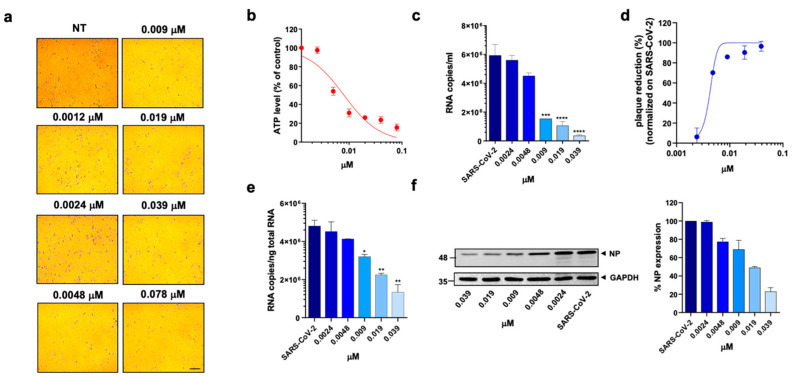
Effect of pralatrexate on A549 ACE2+ cells. A549 ACE2+ cells were cultured for 48 h in the absence or in the presence of PTX at different concentrations. (**a**) 10× bright-field images of A549 ACE2+ cells after incubation for 48 h at 37 °C with the indicated PTX concentration (scale bar, 200 µm). (**b**) CellTiter-Glo was used to measure the antimetabolic effect of PTX. (**c**–**f**) A549 ACE2+ cells were infected with SARS-CoV-2 and cultured in the absence or in the presence of different doses of PTX. (**c**) Viral yield in cell supernatants was quantitated by qRT-PCR. (**d**) Viral titer in cell supernatants was evaluated by plaque assay and plotted as a percentage of plaque reduction compared to SARS-CoV-2. (**e**) Quantitation of SARS-CoV-2 genomes at the intracellular level by qRT-PCR. (**f**) NP expression in infected cells was analyzed by western blot (left panel). Densitometric analysis of western blot, performed by ImageJ software, is shown in the right panel. Graph represents the percentage of NP expression. Data are representative of two independent experiments with similar results. All the experiments were performed at least in three independent replicates, and the pictures shown are representative. Data are presented as the mean ± standard error of the mean; *, *p* < 0.05; **, *p* < 0.01; ***, *p* < 0.001; ****, *p* < 0.0001.

**Figure 3 viruses-15-01128-f003:**
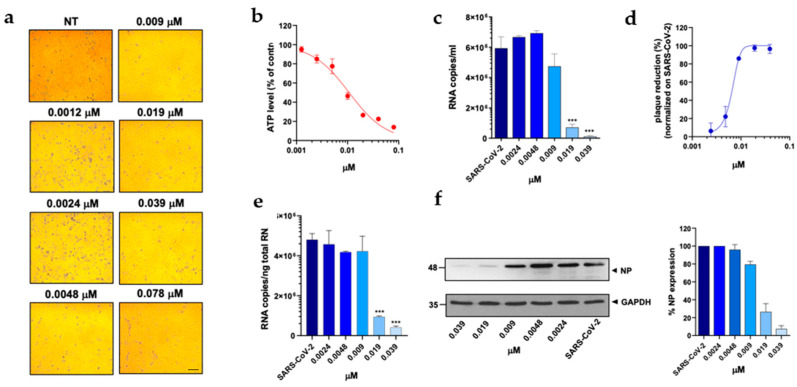
Effect of trimetrexate on A549 ACE2+ cells. A549 ACE2+ cells were cultured for 48 h in the absence or in the presence of TMX at different concentrations. (**a**) 10× bright-field images of A549 ACE2+ cells after incubation for 48 h at 37 °C with the indicated TMX concentrations (scale bar, 200 µm). (**b**) CellTiter-Glo was used to measure the antimetabolic effect of TMX. (**c**–**f**) A549 ACE2+ cells were infected with SARS-CoV-2 and cultured in the absence or in the presence of different doses of TMX. (**c**) Viral yield in cell supernatants was quantitated by qRT-PCR. (**d**) Viral titer in cell supernatants was evaluated by plaque assay and plotted as a percentage of plaque reduction compared to SARS-CoV-2. (**e**) Quantitation of SARS-CoV-2 genomes at the intracellular level by qRT-PCR. (**f**) NP expression in infected cells was analyzed by western blot (left panel). Densitometric analysis of western blot, performed by ImageJ software, is shown in the right panel. Graph represents the percentage of NP expression. Data are representative of two independent experiments with similar results. All the experiments were performed at least in three independent replicates, and the pictures shown are representative. Data are presented as the mean ± standard error of the mean; ***, *p* < 0.001.

**Figure 4 viruses-15-01128-f004:**
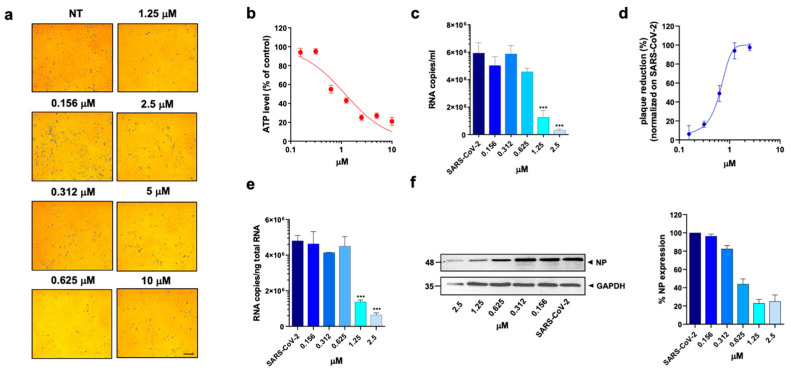
Effect of methotrexate hydrate on A549 ACE2+ cells. A549 ACE2+ cells were cultured for 48 h in the absence or in the presence of MTX at different concentrations. (**a**) 10× bright-field images of A549 ACE2+ cells after incubation for 48 h at 37 °C with the indicated MTX concentrations (scale bar, 200 µm). (**b**) CellTiter-Glo was used to measure the antimetabolic effect of MTX. (**c**–**f**) A549 ACE2+ cells were infected with SARS-CoV-2 and cultured in the absence or in the presence of different doses of MTX. (**c**) Viral yield in cell supernatants was quantitated by qRT-PCR. (**d**) Viral titer in cell supernatants was evaluated by plaque assay and plotted as a percentage of plaque reduction compared to SARS-CoV-2. (**e**) Quantitation of SARS-CoV-2 genomes at the intracellular level by qRT-PCR. (**f**) NP expression in infected cells was analyzed by western blot (left panel). Densitometric analysis of western blot, performed by ImageJ software, is shown in the right panel. Graph represents the percentage of NP expression. Data are representative of two independent experiments with similar results. All the experiments were performed at least in three independent replicates, and the pictures shown are representative. Data are presented as the mean ± standard error of the mean; ***, *p* < 0.001.

**Figure 5 viruses-15-01128-f005:**
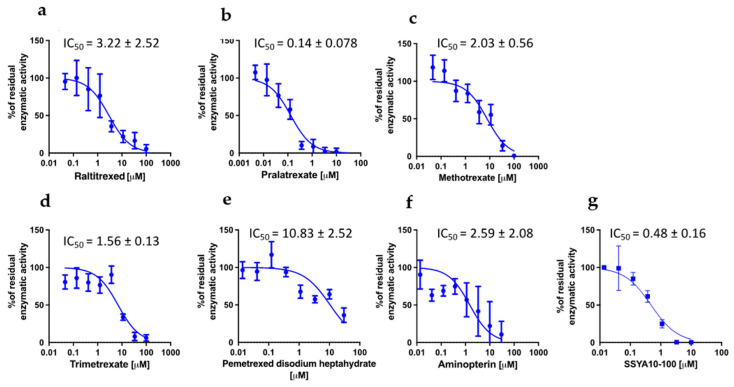
Effect of DHFR inhibitors on nsp13 unwinding activity. Enzymatic assay was performed in the presence and in absence of raltitrexed (**a**), pralatrexate (**b**), methotrexate (**c**), trimetrexate (**d**), pemetrexed disodium heptahydrate (**e**) and aminopterine (**f**). SSYA10-100 was used as a positive control (**g**). Data were collected from two independent experiments in triplicate and presented as the mean ± standard error of residual enzymatic activity.

**Figure 6 viruses-15-01128-f006:**
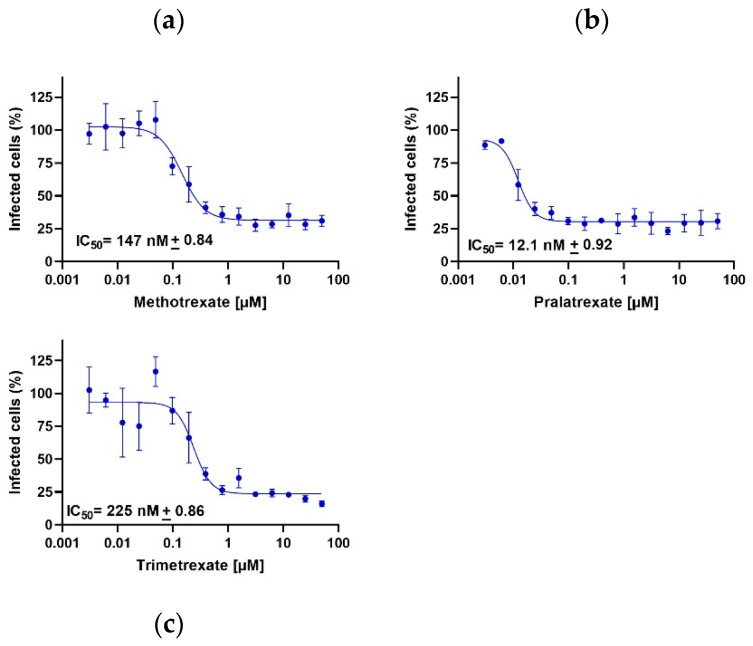
DHFR inhibitors influence the viral entry mechanism. Methotrexate (**a**), pralatrexate (**b**) and trimetrexate (**c**) are active, inhibiting viral infection, in the nanomolar range. Data are presented as the mean ± standard error.

**Table 1 viruses-15-01128-t001:** Anti-viral activity of DHFR inhibitors.

Structures	Drug	IC50 (µM)	CC50 (µM)	SI
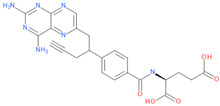	Pralatrexate	0.004	0.008	2
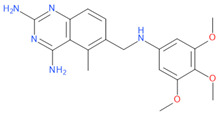	Trimetrexate	0.007	0.01	1.4
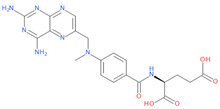	Methotrexate hydrate	0.63	1.18	1.9
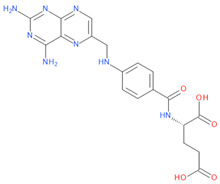	Aminopterin	1.3	2.49	1.9
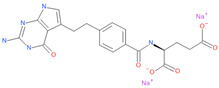	Pemetrexed disodium heptahydrate	5	2.43	0.5
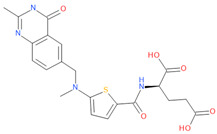	Raltitrexed	5	0.89	0.2

## Data Availability

www.exscalate4cov.eu/results and https://1trilliondock.exscalate4cov.eu/ available from December 2020.

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
