# Peer review of "DHFR Inhibitors Display a Pleiotropic Anti-Viral Activity against SARS-CoV-2: Insights into the Mechanisms of Action"

_viruses, 2023, doi:10.3390/v15051128_

Round 1

Reviewer 1 Report

The Manuscript entitled: DHFR inhibitors display a pleiotropic anti-viral activity against SARS-CoV-2: insights into the mechanisms of action, by Daniela Iaconis and the team, demonstrate the antiviral activity of DHFR inhibitors against SARS-CoV-2. Overall, the manuscript is interesting and informative. I have a few suggestions that I believe will help to improve the overall quality of the manuscript further.  

The manuscript has a few grammatical errors which should be fixed. 

Adding chemical structures of the chemicals tested will be helpful. 

Methods require some elaboration, especially the cells and virus section.  Authors have cited previous references but I encourage authors to briefly describe. 

What was the software used for densitometric analysis?

MTX is patented for anti-viral use from what I found from my search. Authors should consider citing them. 

The table in Figure 1: Cytotoxicity IC50 should be CC50. 

Figure 2 and Figure 3 Image needs scale bars. Authors can also mention that in the legends. 

Figures require more homogeneity in representation. For Example, Figure 5 has +/- and standard error without a unit, unlike Fig. 6.

Figure 6 X-axis should be represented as uM or nM like other figures instead of the log scale. 

If possible, authors should perform a combination experiment of MTX derivatives with known inhibitors. 

Few typographical errors were noticed. For example, 2.5X104 cells/well should be 2.5X10^4

Can be fixed easily.

Reviewer 2 Report

In my opinion, the article is well written, respecting the scientific writing of an original article.

I suggest the authors to change the figure 1 because there are to many graphs in one image, and it is hard to identify the information in each of them. The same for figure 2, 3,4.

Please cite the studies you speak about -Although different studies have sought to explain the effects of MTX using different in 376 vitro models of COVID-19, also suggesting a dual activity of the drug on both viral entry 377 and virus replication in the host, to date the underlying mechanisms of these effects 378 have not been completely explained and targets have not been clearly identified.-

I suggest just a moderate English revision.

Round 2

Reviewer 1 Report

Authors have made changes and the quality of the manuscript is improved. I recommend accepting the manuscript in its current form.